# A Docking Mechanism Based on a Stewart Platform and Its Tracking Control Based on Information Fusion Algorithm

**DOI:** 10.3390/s22030770

**Published:** 2022-01-20

**Authors:** Gan Zhan, Shaohua Niu, Wencai Zhang, Xiaoyan Zhou, Jinhui Pang, Yingchao Li, Jigang Zhan

**Affiliations:** 1School of Mechatronical Engineering of Beijing Institute of Technology, Beijing 100081, China; 3120160145@bit.edu.cn (G.Z.); 3220200178@bit.edu.cn (W.Z.); 2School of Electrical Engineering, Beijing Jiaotong University, Beijing 100044, China; 20121543@bjtu.edu.cn; 3School of Computer Science and Technology of Beijing Institute of Technology, Beijing 100081, China; pangjinhui@bit.edu.cn; 4Beijing Zhongxin Hengyuan Technology Co., Ltd., Beijing 100081, China; zhonghangzhuoyuan@163.com (Y.L.); zg18813167598@163.com (J.Z.)

**Keywords:** docking mechanism, Stewart platform, visual control, fusion algorithm

## Abstract

Aiming at the problem of unmanned reconfiguration and docking of ground vehicles under complex working conditions, we designed a piece of docking equipment composed of an active mechanism based on a six-degree-of-freedom platform and a locking mechanism with multi-sensors. Through the proposed control method based on laser and image sensor information fusion calculation, the six-dimensional posture information of the mechanism during the docking process is captured in real time so as to achieve high-precision docking. Finally, the effectiveness of the method and the feasibility of the 6-DOF platform are verified by the established model. The results show that the mechanism can meet the requirements of smooth docking of ground unmanned vehicles.

## 1. Introduction

The concept of intelligent vehicles puts forward new requirements for vehicle formation, reconstruction and docking. At the same time, with the popularity of the concept of driverless driving, the need to combine unmanned ground vehicles with autonomous reconstruction and docking has also become the focus of research in the field of intelligent robots and machinery. Among them, the docking technology of robots, as the basic technical support, was initially used in scenarios such as autonomous charging of robots, data transmission and space attitude capture [1,2,3,4] and has now been widely used in aircraft cross-linking docking, aerial refueling and ocean exploration submarine docking in aviation [5,6,7,8,9,10]. This technology, which can establish physical connections autonomously, provides a new idea for the current ground vehicle formation, which needs to use high-performance computers and rely on real-time control communication systems to realize complex computation to carry out information interaction.

The 6-DOF motion platform, also known as the Stewart platform, is a typical parallel robot for flight simulators proposed by the scholar Stewart in 1965 [11], which provides good hardware support for the realization of robot docking technology and has been widely recognized. As the hardware equipment of the docking mechanism, the 6-DOF parallel motion platform is widely recognized at present. The advantages of the parallel structure, such as large relative stiffness, small inertia of the mechanism, precise position, high system bandwidth and strong relative load capacity, are irreplaceable for the series mechanism [12,13,14]. Therefore, the reconfiguration of docking technology based on the 6-DOF motion platform has been widely used in various carrier simulators, launch vehicle connector automatic docking, multi-degree-of-freedom parallel mechanical equipment, ultra-precision micro-nanometer systems and mobile robot equipment [15,16,17,18,19], which provides a technical and theoretical reference for the reconfiguration docking of ground unmanned vehicles. For example, a novel six-wheel-legged robot based on a 6-DOF parallel device designed by Junzheng Wang’s team [20,21,22,23,24,25] realizes multi-modal motion. Javier Velasco [26] improved the performance and accuracy of a positioning system based on the Stewart platform by proposing a sliding mode control strategy for accurate positioning, and this control scheme provides a reference for application to larger motion platforms and complex simulation robots. In the research and development of a driving simulator, Yash Raj Khusro [27] designed a 6-DOF simulator with actuator constraints based on the MPC motion hint algorithm, which improved the accuracy of the 6-DOF motion platform under nonlinear motion conditions. Cosmin Copot [28] designed a 6-DOF motion platform for realizing virtual environment space docking operation equipment under manual monitoring. Sang-moon Yun [29] designed the 6-DOF platform equipment for the shaking experiment of sea-surface transport vessels and conducted a safety assessment on the connection between the shaking tank and the 6-DOF platform. Alaa Eldin Abdelaa [30] designed a new method of autonomous camera movement with 6-DOF for minimally invasive surgery. The above expansion and application of a 6-DOF motion platform in the field of docking mechanisms provides theoretical and technical reference for the realization of unmanned ground vehicle reconstruction docking.

Among them, the ground vehicle achieves unmanned reconstruction and docking; the requirements for the structure of the control system and the hardware platform are as simple as possible; and the locking mechanism is connected to the active mechanism in the complex conditions [31]. In references [32,33,34,35,36], nonlinear problems such as space electromagnetic docking are studied. Feedback linearization and internal and external loop control strategies are applied to make the docking process smoother. However, in the docking process, the acquisition of the position and attitude of the docking mechanism also needs the help of external auxiliary devices other than the six-degree-of-freedom platform. The position and attitude information of the mechanism captured by the auxiliary device also determines the accuracy of the six-degree-of-freedom motion platform for vehicle docking. In autonomous vehicle technology, the sensor network and sensor fusion system are designed to accurately draw the surrounding environment information learned by safe driving. This method is also suitable for information collection of the docking mechanism under complex conditions on the ground to achieve accurate docking. Javeria Khan [37] proposed a brain-driven intelligent vehicle with sonar and visual sensor networks and controlled the joystick through steady-state visual evoked potential brain signals. In order to compensate for the error caused by sensor limitations, Akshaya T.G. [38] used the sensor fusion algorithm of the Kalman filter to enable ground vehicles to obtain continuous error information from GPS. Yang Yuan and Huang Yongjiang [39] used the multi-sensor fusion algorithm of Global Navigation Satellite System (GNSS), Lidar-Lite (laser rangefinder equipment), barometer and low-cost inertial navigation system (INS) to control and track the quadrotor aircraft in real time so that it can respond to real-time continuous navigation and obstacle avoidance. Studying the multi-sensor data fusion algorithm, Setareh Yazdkhasti [40] proposed an integration algorithm based on the combination of fuzzy logic controller and Kalman filter, which further enhanced the reliability and accuracy of data. The above multi-sensor information fusion algorithm control strategy applied to driving vehicles also provides technical support for the reconstruction and docking of unmanned ground vehicles.

The work of this paper is to design a docking mechanism based on a 6-DOF motion platform and use the information fusion algorithm of laser and visual sensors to realize the autonomous reconfiguration and docking of unmanned vehicles under complex ground conditions. The main contributions of this paper are as follows:

(1) A simulation model of a docking mechanism based on a 6-DOF motion platform as the active mechanism and locking mechanism with external auxiliary equipment is established to realize vehicle unmanned docking and reconstruction under complex ground conditions.

(2) In the six-dimensional posture information measurement of the mechanism docking process, a control method based on information fusion of laser and image sensors is proposed. The position and posture information are captured by the laser ranging sensor installed on the active mechanism, and the locking mechanism and target recognition camera are weighted and fused to improve the docking accuracy of the mechanism.

The content structure of this paper is as follows: In the second section, the system architecture and working principle of the docking mechanism are given. In the third section, the principle of the six-dimensional pose fusion algorithm based on laser and image sensors is explained. In the fourth section, the docking system and the two-sensor information fusion control algorithm are verified by simulation and experiment. Finally, a summary is made, and the future prospects for the further optimization of the mechanism and algorithm are put forward.

## 2. Docking Mechanism Based on Six-Degree-of-Freedom Motion Platform

In the reorganization of the docking mechanism of unmanned vehicles applied to the ground, it is very difficult to achieve dynamic rigid docking due to the influence of complex factors such as the terrain environment. This paper mainly solves the problems of vision technology such as the capture of environmental information and target recognition and measurement of the organization during dynamic docking. By adding a target recognition camera and a laser ranging sensor to the conventional docking mechanism, the pose states of the two mechanisms are captured in real time, and the weighted fusion vision control algorithm is used to achieve precise docking of the platform in a dynamic environment. This section introduces the docking mechanism model and the docking process of the mechanism using the fusion control method.

### 2.1. Docking Mechanism Model

The docking mechanism is composed of a 6-DOF active mechanism and a locking mechanism. The active mechanism is fixed at the front of the car and the passive mechanism is fixed at the rear of the vehicle. When docking, the position of the docking block is adjusted by the 6-DOF platform, and the docking block is inserted into the locking mechanism to lock the inside of the mechanism. The pin shaft extends and locks with the docking block to realize a rigid connection. The effect picture of the docking mechanism applied to different car bodies to achieve docking tasks is shown in Figure 1 below.

Among them, the active mechanism is mainly composed of the following components: upper platform, electric cylinder, lower platform base, reducer servo motor, servo drive system, Hooke articulation, attitude computing system, control system, recognition positioning camera, laser ranging sensor, etc. The components of the active mechanism are shown in Figure 2a. The locking mechanism mainly includes: image recognition positioning plate, radial locking pin, two laser sensors, laser sensor detection plate, triangular claw plate, torque motor and guide rod. Its components are shown in Figure 2b.

The design idea of the mechanism using vision for docking is to install a recognition positioning camera on the active mechanism. The camera recognizes the image recognition positioning plate installed on the locking mechanism to achieve target capture. The positioning plate is composed of three RGB primary color pixel blocks, which helps the camera capture. The plane composed of the color segmentation vertices in the middle of the color block is used as the target image, and the above-mentioned image is projected into the camera coordinate system for calibration through the camera external parameter matrix transformation. At the same time, five laser ranging sensors have been added. Three laser ranging sensors installed on the active mechanism work together with the laser detection board of the locking mechanism to realize the real-time capture of the poses of the two platforms when the active mechanism advances. The two are installed on the locking mechanism and are used to trigger the advance and stop actions of the locking mechanism and then determine whether the docking is in place. The two methods of camera and laser sensor jointly control the precise docking of the docking mechanism under dynamic conditions through the control algorithm of weighted fusion. The visual positioning schematic diagram of the docking system is shown in Figure 3 below.

### 2.2. Working Principle of Docking Mechanism 

The workflow of the docking mechanism is as follows:(A)After the power is turned on, the 6-DOF platform system is initialized first, and the target recognition camera and laser ranging sensor start to work. The working space of the moving platform is w, and the locking mechanism is always in the docking area W. First, the laser ranging sensor judges whether the moving platform is in the docking area W (w∈W). If it is not, you need to adjust the two cars directly. The position is relative; if it is in the docking area, start immediately.(B)First, the controller controls the extension of the six electric cylinders to drive the active mechanism forward to the neutral position, so that the position error between the active mechanism and the locking mechanism of the six-axis platform is gradually reduced. During this period, the recognition and positioning camera and the laser ranging sensor located in the active mechanism perform target capture and real-time feedback of the pose status and transmit the information to the controller, and the six-axis platform locks the docking center for position control.(C)When it is recognized that the distance between the platform and the locking mechanism is less than a certain threshold, the locking mechanism starts to move, and the docking push rod pushes the platform forward. When the distance between the two detected by the laser ranging sensor is constant, the triangular claw is locked tightly for the first time, and the radial locking pin located in the locking mechanism is controlled by a small electric cylinder to drive the spring pin assembly to spring into the pin hole.(D)At this time, observe whether there is a change in the value of the pressure sensor. If there is any change, there is still an error in the relative position of the two mechanisms. You need to rotate the docking structure until the value of the pressure sensor is zero; now it should be successfully docked. The docking flow chart is shown in Figure 4 below.

## 3. Control Algorithm of Docking Mechanism Based on Image and Laser Sensor Fusion

The system first controls the 6-DOF platform and then transmits the relative position difference between the identification and positioning camera and the calibration plate to the main controller. The main controller can use this parameter and use negative feedback to control the 6-DOF platform in combination with the laser ranging sensor to realize docking.

Firstly, docking establishes the transformation relationship of each coordinate system, in which the coordinate systems of the moving platform and the static platform are $O-X_{A}Y_{A}Z_{A}$, $O-X_{B}Y_{B}Z_{B}$, and the static platform is fixedly connected with the world coordinate system $O-X_{W}Y_{W}Z_{W}$. The camera coordinate system, pixel coordinate system and laser coordinate system are $O-X_{C}Y_{C}Z_{C}$, $O-X_{S}Y_{S}Z_{S}$, $O-X_{L}Y_{L}Z_{L}$, respectively, and the transformation diagram is shown in Figure 5.

### 3.1. Inverse Solution Algorithm

In order to facilitate the detailed kinematic analysis of the Stewart parallel mechanism, the mathematical model shown in Figure 6 is established. The 6-DOF platform is composed of an upper platform (dynamic platform) and lower platform (static platform).

The attitude angle of the moving platform equivalent to the static platform is α,β,γ. Suppose that the coordinates of the origin $O_{A}$ of the moving platform in the coordinate system of the static platform are $A_{O}^{B}={\left[\begin{array}{ccc} x & y & z \end{array}\right]}^{T}$, the coordinates of the points on the moving platform relative to the moving platform are $A_{i}^{A}=\left[\begin{array}{ccc} x^{\prime } & y^{\prime } & z^{\prime } \end{array}\right]$, the coordinates of the points on the static platform relative to the static platform are $B_{i}^{B}=\left[\begin{array}{ccc} x^{\prime \prime } & y^{\prime \prime } & z^{\prime \prime } \end{array}\right]$, and the expression of the rotating torque matrix of the rotating platform equivalent to the rotating attitude of the static platform is as follows:(1)$$T=\left[\begin{array}{ccc} \cos\beta \times \cos\gamma & \cos\beta \times \sin\gamma \times \sin\alpha -\sin\gamma \times \cos\alpha & \sin\gamma \times \sin\alpha +\cos\gamma \times \sin\beta \times \cos\beta \\ \cos\beta \times \sin\gamma & \cos\gamma \times \cos\alpha +\sin\gamma \times \sin\beta \times \sin\alpha & \sin\gamma \times \sin\beta \times \cos\alpha -\cos\gamma \times \sin\alpha \\ -\sin\beta & \cos\beta \times \sin\alpha & \cos\beta \times \cos\alpha \end{array}\right]$$

Then the coordinates of the points on the moving platform relative to the static platform can be expressed as follows:(2)$$A_{i}^{B}=T\times A_{i}^{A}+A_{o}^{B}$$

The expression of elongation of $l_{i}$ each cylinder can be expressed as:(3)$$\left|l_{i}\right|=\left|A_{i}^{B}-B_{i}^{B}\right|$$

### 3.2. Multi-Sensor Information Fusion Algorithm for Image and Laser

#### 3.2.1. Image Perception Algorithm

To obtain the visual control algorithm, first calibrate the camera. The calibration principle is shown in Figure 7:

The camera calibration is divided into two steps:The calibration board is transformed from pixel coordinate system to camera coordinate system through the camera internal parameter matrix;The calibration board is transformed from the camera coordinate system to the world coordinate system through the camera external parameter matrix.

The position of the target in the pixel coordinate system $P{\left(\begin{array}{ccc} X_{S} & Y_{S} & Z_{S} \end{array}\right)}^{T}$ is converted to the position of D in the camera coordinate system through the focal length diagonal matrix and pixel conversion matrix $P{\left(\begin{array}{ccc} X_{C} & Y_{C} & Z_{C} \end{array}\right)}^{T}$, which can be expressed as:(4)$$\left[\begin{array}{c} X_{C}\\ Y_{C}\\ Z_{C} \end{array}\right]={\left[\begin{array}{ccc} f & 0 & 0\\ 0 & f & 0\\ 0 & 0 & 1 \end{array}\right]}^{-1}\times {\left[\begin{array}{ccc} \frac{1}{\mathrm{dx}} & 0 & u_{0}\\ 0 & \frac{1}{\mathrm{dy}} & v_{0}\\ 0 & 0 & 1 \end{array}\right]}^{-1}\times \left[\begin{array}{c} X_{S}\\ Y_{S}\\ Z_{S} \end{array}\right]$$
where dx,dy represent the unit pixel length in x,y direction; $\left(u_{0}v_{0}\right)$ represents the origin of the pixel coordinate system; and f is the focal length. 

The position of the object in the world coordinate system is set as $P{\left(\begin{array}{ccc} X_{W} & Y_{W} & Z_{W} \end{array}\right)}^{T}$, and through the external parameter matrix in the camera coordinate system, it can be expressed as Equation (5):(5)$$\left[\begin{array}{c} X_{W}\\ Y_{W}\\ Z_{W} \end{array}\right]={\left[\begin{array}{cc} R & T\\ 0 & 1 \end{array}\right]}^{-1}\times \left[\begin{array}{c} X_{C}\\ Y_{C}\\ Z_{C} \end{array}\right]$$
where R is the rotation matrix and T is the translation matrix. Thus, the three-pose information of $\left[\begin{array}{ccc} X_{W} & Y_{W} & Z_{W} \end{array}\right]$ can be obtained. Suppose the projection of the calibration plate in the pixel coordinate system is shown in Figure 8 below.

Assuming |UD|=|LR|=lan, the corresponding pose information can be obtained from the spatial geometric relationship:(6)$$\mathrm{Yaw}\;\mathrm{Angle}:\;\alpha =\arctan\frac{u_{y}}{u_{x}}$$
(7)$$\mathrm{Pitching}\;\mathrm{Angle}:\;\beta =\arccos\sqrt{\frac{\left(u_{y}{}^{2}+u_{x}{}^{2}\right)}{\mathrm{lan}}}$$
(8)$$\mathrm{Roll}\;\mathrm{Angle}:\;\gamma =\arccos\frac{\left(u_{y}\times l_{x}-u_{x}\times l_{y}\right)}{(\mathrm{lan}\times \sqrt{\left(u_{y}^{2}+u_{x}^{2}\right)}}$$
where the projection plane $L_{0}U_{0}R_{0}D_{0}$ spatial position coordinates are vector $L_{0}R_{0}=(l_{x},l_{y},0)$, vector $U_{0}D_{0}=(u_{x},u_{y},0)$, vector $\mathrm{LR}=(l_{x},l_{y},p)$ and vector $\mathrm{UD}=(u_{x},u_{y},q)$; p,q is unknown. 

Through the above calculation method, the actual spatial position of the target image can be obtained, and the obtained position information can be sent to the main controller in real time. The main controller performs relevant operations and makes real-time position feedback. The docking attitude detection scheme based on the ranging sensor is mainly to adjust the attitude of unmanned vehicles and realize the parallelism of front vehicles and rear vehicles. In this scheme, the deviation of the locking mechanism relative to the six-degree-of-freedom coordinate system can be fed back in real time so as to provide the control command of the six-degree-of-freedom module and realize the parallelism of the front vehicle and the rear vehicle.

#### 3.2.2. Laser Measurement Algorithm

The laser ranging sensor adopts the three-point method to detect the attitude of the six-degree-of-freedom mechanism. Its basic principle is shown in Figure 9:

The laser emission points of the three laser ranging sensors are located on the three vertices of the equilateral triangle ABC. The laser beams are parallel to each other and perpendicular to the plane ABC, assuming that the plane S is the measured object surface and the laser irradiation point is $A_{1},B_{1},C_{1}$, so the distance from the three points to the plane is $L_{A}=\left\|AA_{1}\right\|,L_{B}=\left\|AA_{2}\right\|,L_{C}=\left\|AA_{3}\right\|$. 

Using rotation matrix and translation matrix $\left[\begin{array}{cc} R & T\\ 0 & 1 \end{array}\right]$ can express θ and ϕ by Equation (9):(9)$$R=\left[\begin{array}{cccc} \cos\phi & 0 & \sin\phi & 0\\ \sin\theta \sin\phi & \cos\theta & -\sin\theta \cos\phi & 0\\ -\cos\theta \sin\phi & \sin\theta & \cos\theta \cos\phi & L_{A}\\ 0 & 0 & 0 & 1 \end{array}\right]$$

Bring $L_{A},L_{B},L_{C}$ into the transformation matrix to obtain:(10)$$\begin{array}{l} \theta =\arcsin(\frac{L_{B}-L_{c}}{\mathrm{BC}})\\ f=\arcsin(\frac{L_{B}+L_{C}-L_{A}}{O_{1}D})\\ Z=L_{A} \end{array}$$

#### 3.2.3. Fusion Algorithm of Image and Laser Sensor Information

The method of image and laser sensor information fusion is adopted to facilitate the six-degree-of-freedom platform to properly adjust the attitude of the docking platform and make corresponding control instructions to the six-degree-of-freedom platform in real time. The specific control methods are as follows: x,y,z,α,β,γ is the information of the six degrees of freedom, and assuming that the pose information obtained by the camera is M[x,y,z,α,β,γ] and the pose information obtained by the three laser sensors is $N[z_{1},\theta ,f]$, the optimal pose information E[x,y,z,χ,η,ν] can be expressed as:(11)$$E\left[x,y,z,\chi ,\eta ,\nu \right]=M\left[x,y,z,\alpha ,\beta ,\gamma \right]\times \lambda +N\left[z_{1},\theta ,f\right]\times \mu$$
where α is the pitch angle, β is the yaw angle and γ is the roll angle; 0≤μ≤1,0≤λ≤1,μ+λ=1.

The information obtained based on the image can be fully weighted with the information obtained based on the laser sensor, so that the position information can be obtained more accurately.

### 3.3. Control Algorithm

The six-dimensional attitude information obtained by the laser and vision sensors for the locking mechanism is obtained based on the moving platform and transmitted to the task controller. The PC will get the command, calculate the position and attitude of the static platform relative to the locking mechanism and then send the command. The calculation process is as follows: the pose command of the static platform relative to the locking mechanism can be expressed as
(12)$$\left\{ \begin{array}{l} x=-x_{1}\\ y=-y_{1}\\ z=-z_{1}\\ \alpha =\alpha_{1}+\alpha_{2}\\ \beta =\beta_{1}+\beta_{2}\\ \gamma =\gamma_{1}+\gamma_{2} \end{array}\right.$$
where it is assumed that the pose information is (x1,y1,z1,α1,β1,γ1); the pose of the moving platform relative to the static platform is (x2,y2,z2,α2,β2,γ2).

## 4. Simulation and Results

### 4.1. Model

For the solution of model establishment, we use SolidWorks to model the docking mechanism, as shown in Figure 10 below. The relevant parameters of the 6-DOF platform are shown in Table 1, and the parameters of the electronic system are shown in Table 2.

The established model is imported into MATLAB, and the moving pair and rotating pair are added in Simulink. The relative pose between coordinate systems is used to simulate the observation of vision and laser sensors, and noise interference is added to simulate the actual working condition. The simulation diagrams of Simulink are shown in Figure A1 in the Appendix A.

### 4.2. Simulation

The kinematics simulation of the system model is carried out in the simulation software. Firstly, move the platform 75 mm in the z-axis direction and then rotate 15° around the *y*-axis. In this process, we vibrate the locking mechanism through external interference (i.e., random square wave noise signal), which is used to simulate the visual sensor on the road due to measurement error by bumps (α1,β1,γ1). The measurement error of the visual sensor and under this influence is not very obvious, but the laser sensor only measures the three-dimensional attitude and cannot get the distance information (x, y). Therefore, we use a combination of laser sensors and visual sensors. The optimal attitude solution is obtained by Kalman filtering. After the platform is finally modified, the locking mechanism advances to complete the docking.

In the whole process, when T = 1.4 s and T = 2.6 s, we apply an interference signal (random square wave) to the locking mechanism, and the active mechanism also vibrates with the locking mechanism. The vision sensor on the mobile platform measures the six-dimensional attitude relative to the locking mechanism, in which the Euler angle is obtained by quaternion(α1,β1,γ1); the distance information (x, y, z) and quaternion curve are shown in Figure 11 below.

Through the following Equation (13), the Euler angle can be obtained as shown in Figure 12:(13)$$\begin{array}{l} \alpha_{1}=\arctan\left(\frac{2(q_{0}q_{1}+q_{2}q_{3})}{1-2(q_{1}^{2}+q_{2}^{2})}\right)\\ \beta_{1}=\arcsin(2(q_{0}q_{2}-q_{1}q_{3}))\\ \gamma_{1}=\arctan\left(\frac{2(q_{0}q_{3}+q_{2}q_{1})}{1-2(q_{2}^{2}+q_{3}^{2})}\right) \end{array}$$

It can be clearly seen from Figure 11 and Figure 12 that when the locking mechanism jitters, the visual sensor also jitters, resulting in measurement error. The vision sensor produces an error of 2.5° when the platform actually rotates 15°. The other two attitude angles have relative errors, but γ1 shows little change, and α1 also produces an error of 2.5°. At this time, the laser sensor is used for weighted filter processing, and the three laser sensors on the moving platform are used to measure relative to the lock. The distance information of the organization is shown in Figure 13 below.

Through the data obtained by the three sensors, it can be observed that the rotation trend of the moving platform is correct. We bring the data measured in the above figure into the Chapter 3 Laser Sensor Algorithm to obtain the attitude solution of the platform and use this data to perform Kalman filtering on the attitude measured by the vision sensor. The result is shown in Figure 14 below.

As shown in the figure above, the filtered curve is the optimal result of vision and laser sensor fusion. It can be seen from Figure 14a that the optimal result depends more on the data measured by the laser sensor, indicating that the data of the visual sensor has a large error due to vibration at this time. It can be seen from Figure 14b that the optimal result depends on the measured results of both laser sensors and visual sensors, indicating that there are errors in both sensors at this time. The optimal result is obtained after filtering according to the measured results of the previous group and the data obtained by visual and laser sensors. 

At this time, only the data measured by the visual sensor and the optimal result after Kalman filtering will be used to track the locking mechanism applying an interference signal (random square wave signal), as shown in Figure 15 below.

It can be clearly observed from the above figure that the tracking effect of the vision and laser fusion algorithm is significantly better than that of only using visual sensors. The tracking accuracy is greatly improved. The Kalman filtered data is fed back to the controller, and after t = 2 s, the position of the moving platform is adjusted until it is coaxial with the locking mechanism and the locking mechanism is triggered to advance to complete the docking. During the whole process, the measured pose curve (quaternion vs. distance curve) of the moving platform relative to the static platform is shown in Figure 16 below.

By comparing the attitude of the moving platform relative to the static platform obtained in Figure 16 above with that of the moving platform relative to the locking mechanism in Figure 11, the attitude command of the static platform relative to the locking mechanism can be easily obtained through Equation (12), so as to control the moving platform to adjust the attitude and realize docking. The position curves of the six electric cylinders all along the line are shown in Figure 17 below.

As shown in Figure 18 below, the force curve of the force sensor installed at the end of six electric cylinders during docking is shown. It can be seen that when t = 0.5 s, the first collision between the active mechanism and the locking mechanism occurred on the side of No. 5 electric cylinder (light blue curve in the figure). Then, when t = 0.6 s, the force curves of No. 1, 2, 3 and 6 electric cylinders changed significantly, indicating that the locking mechanism collided completely with the active mechanism at this time, which means that the locking mechanism has penetrated into the active mechanism and realized pre docking.

## 5. Conclusions

In this paper, a docking mechanism for ground vehicle reconfiguration and docking is designed. On the active mechanism, a six-degree-of-freedom motion platform with excellent performance is adopted.On the locking mechanism, by adding external equipment—target recognition, positioning camera and laser ranging sensor—and using the control method of laser and visual sensor information fusion, the vehicle can realize smooth docking under complex ground conditions and have high docking accuracy. In the final experimental verification, it can be seen that under the condition of applying interference noise signals to simulate bumpy conditions, the position and posture curves measured by the moving platform are basically consistent with the position and posture curves of the locking mechanism. This clearly shows that under the visual fusion control method based on camera and laser, the active mechanism has good tracking performance during locking and docking and can complete the smooth docking of the two mechanisms.

At present, the work done in this paper only processes the weighted fusion information of the sensors added with two-pose information measurements and simulates the follow-up tracking of the locking mechanism to the active mechanism in the bumpy state, while the reconstruction and docking environment of ground vehicles is more complex and diverse. Therefore, it is also necessary to set more complex dynamic conditions, consider adding six-dimensional force sensors, optimize the control method of information fusion, and simulate the working conditions that may be faced during actual ground docking from six dimensions, which is also the focus of our next research.

## Figures and Tables

**Figure 1 sensors-22-00770-f001:**
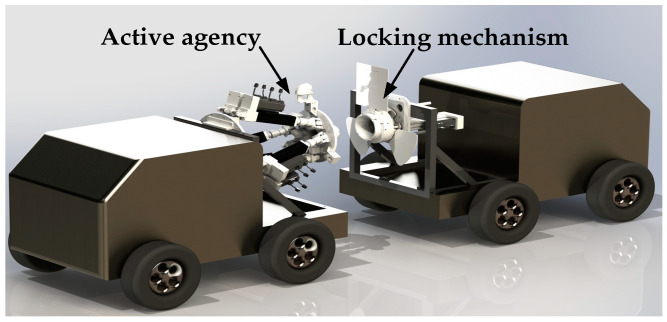
A rendering of the reconfiguration docking of an unmanned vehicle.

**Figure 2 sensors-22-00770-f002:**
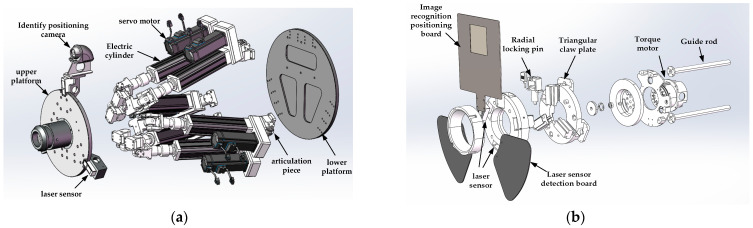
An exploded view of the docking mechanism, where (**a**) is an exploded view of the active mechanism and (**b**) is an exploded view of the locking mechanism.

**Figure 3 sensors-22-00770-f003:**
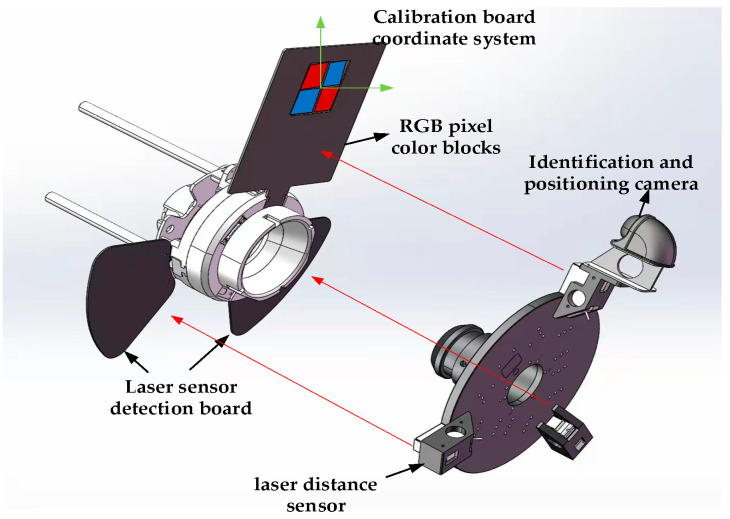
Schematic diagram of visual positioning of the docking system.

**Figure 4 sensors-22-00770-f004:**
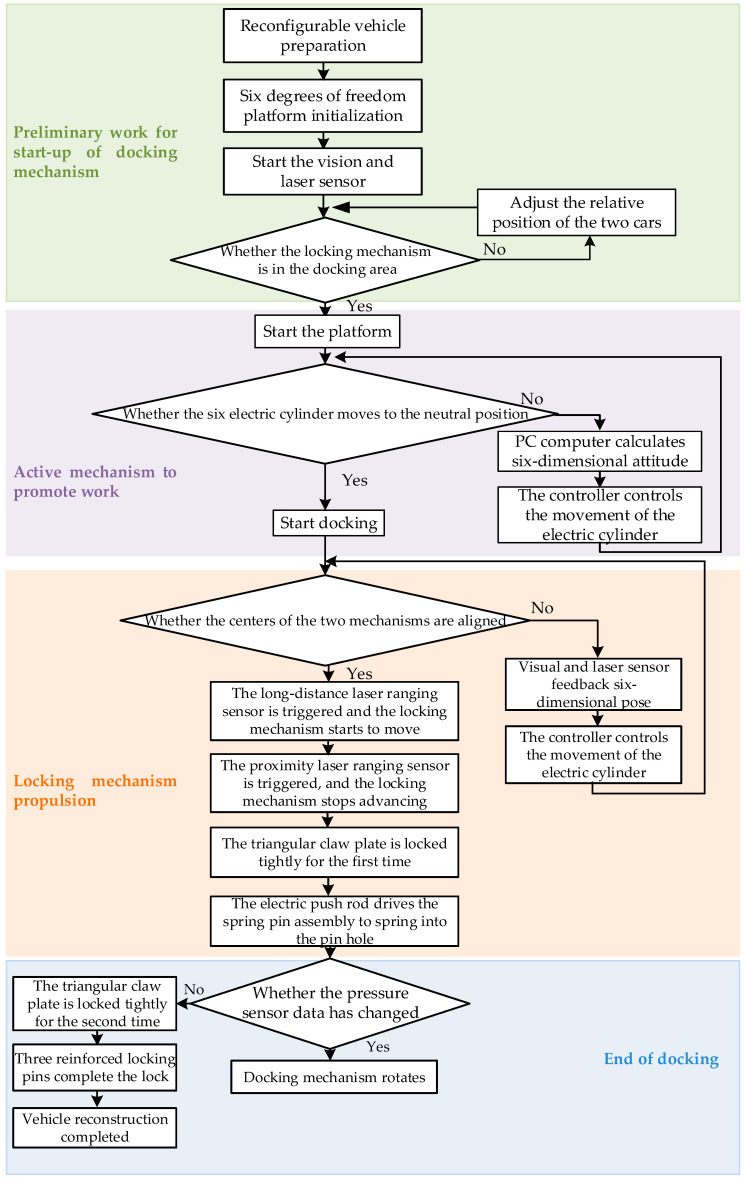
Flow chart of docking process.

**Figure 5 sensors-22-00770-f005:**
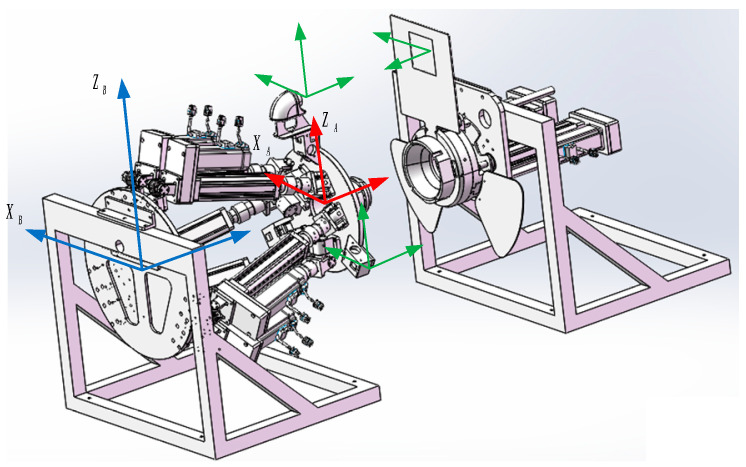
Transformation diagram of the docking system coordinate system.

**Figure 6 sensors-22-00770-f006:**
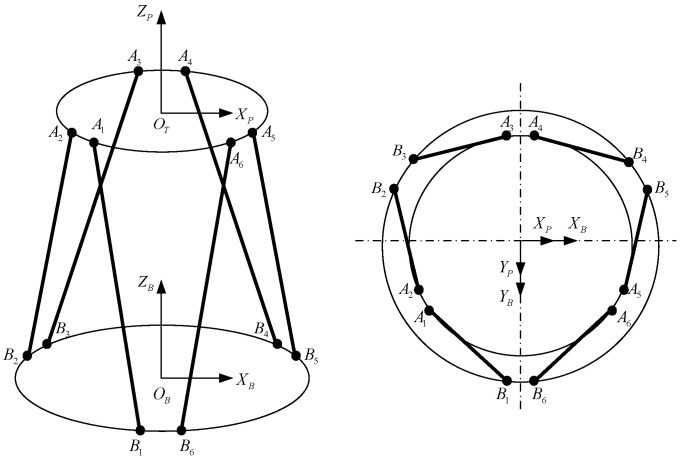
Simplified mathematical model of Stewart parallel mechanism.

**Figure 7 sensors-22-00770-f007:**
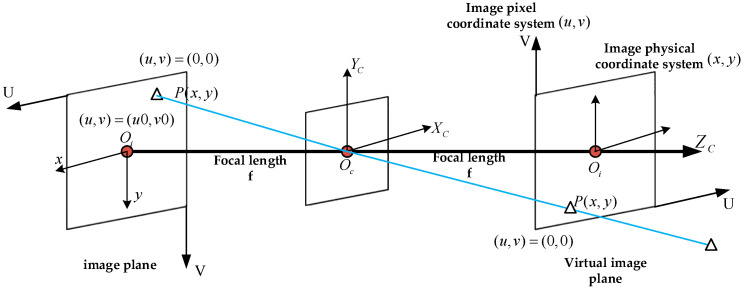
Calibration schematic diagram.

**Figure 8 sensors-22-00770-f008:**
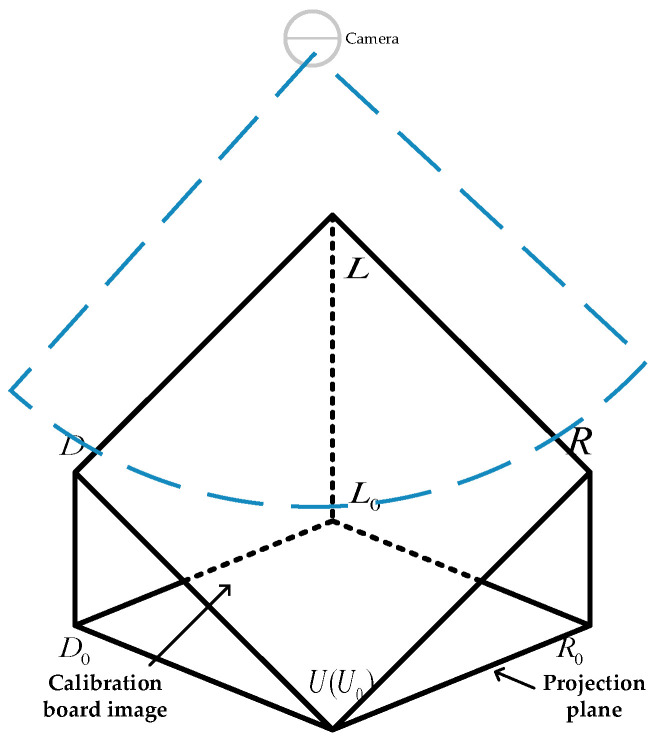
Spatial projection.

**Figure 9 sensors-22-00770-f009:**
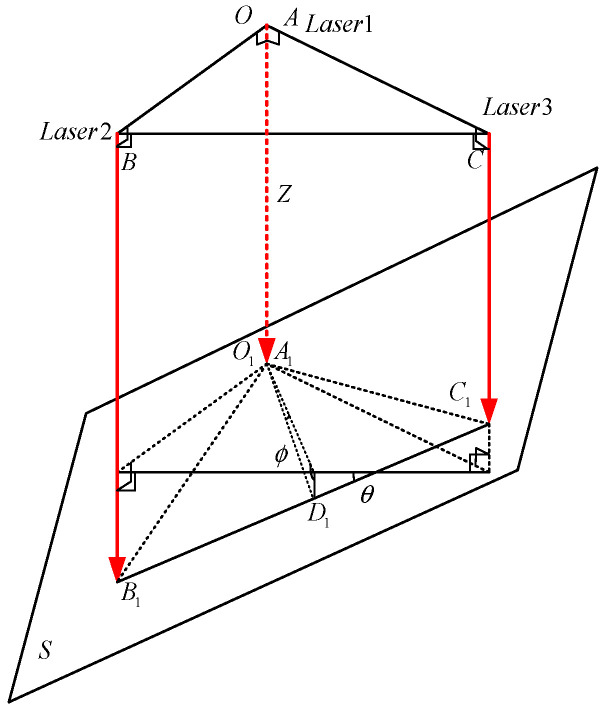
Schematic diagram of three-point method.

**Figure 10 sensors-22-00770-f010:**
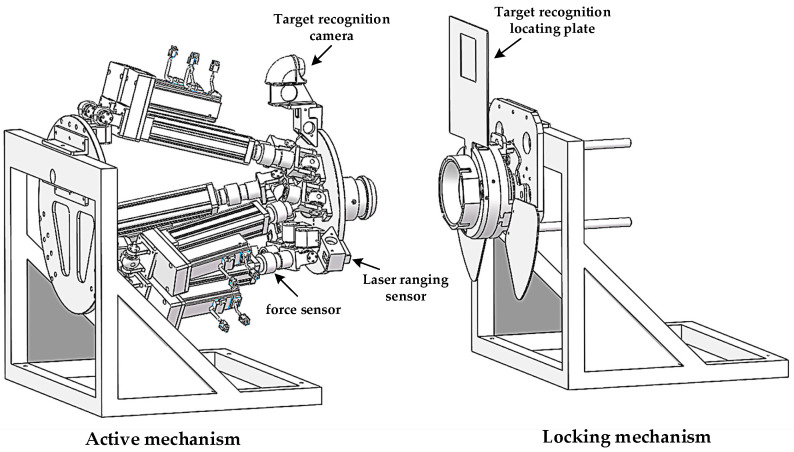
Hardware distribution diagram of docking mechanism.

**Figure 11 sensors-22-00770-f011:**
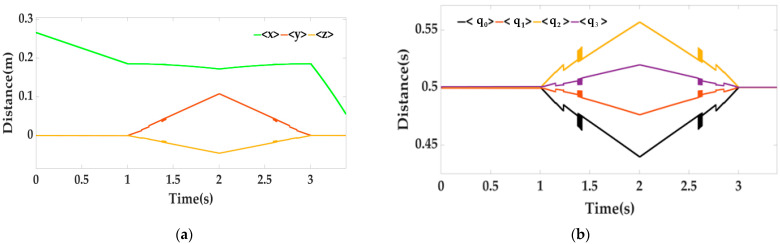
Six-dimensional posture graph, where (**a**) is the distance information graph and (**b**) is the quaternion graph.

**Figure 12 sensors-22-00770-f012:**
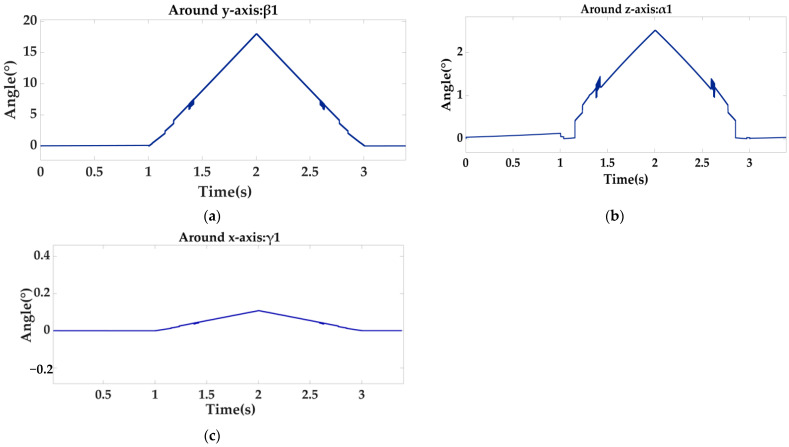
Euler angle diagram measured by vision sensor, rotated 15° around the y-axis, where (**a**) is β_1_; (**b**) is α_1_; (**c**) is γ_1_.

**Figure 13 sensors-22-00770-f013:**
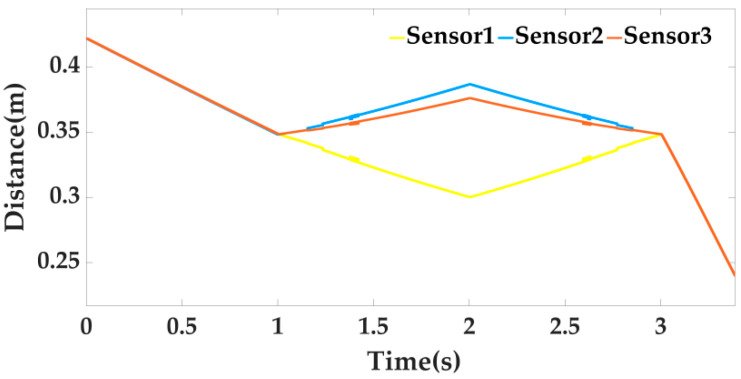
Distance of three laser sensors.

**Figure 14 sensors-22-00770-f014:**
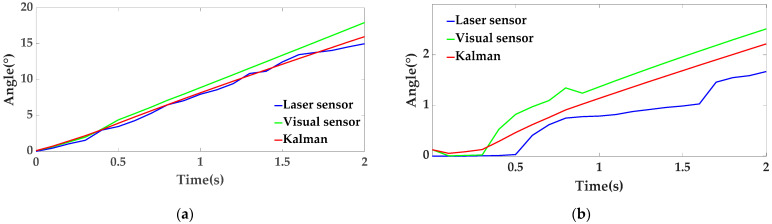
Euler angle diagram measured by vision sensor, laser sensor and Kalman, where (**a**) represents the Kalman filter analysis for β_1_ and (**b**) represents the Kalman filter analysis for α_1_.

**Figure 15 sensors-22-00770-f015:**
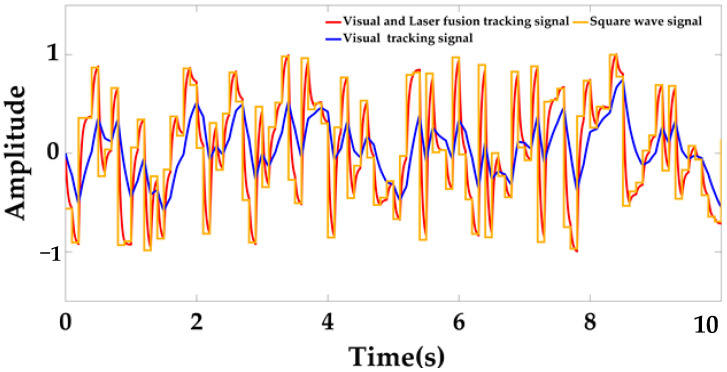
Comparison diagram of tracking effect.

**Figure 16 sensors-22-00770-f016:**
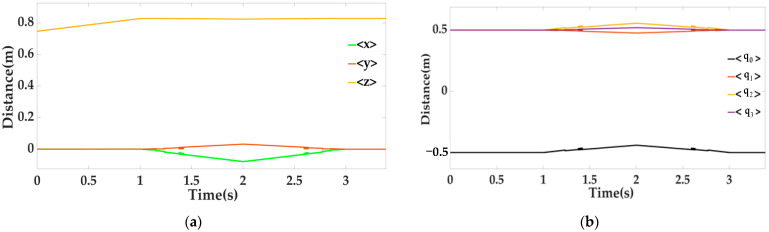
The pose curve of the moving platform relative to the static platform, where (**a**) is the quaternion curve diagram and (**b**) is the distance curve diagram.

**Figure 17 sensors-22-00770-f017:**
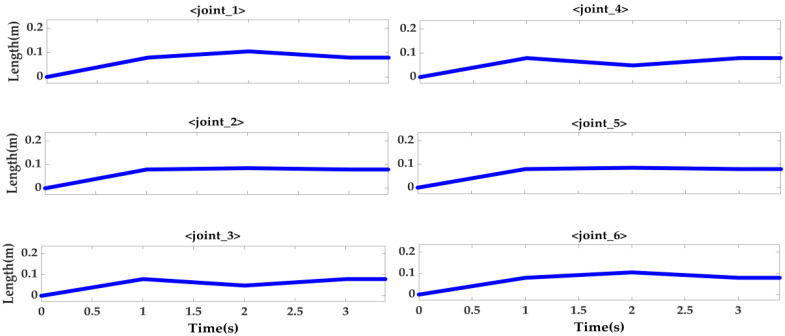
Electric cylinder displacement curve.

**Figure 18 sensors-22-00770-f018:**
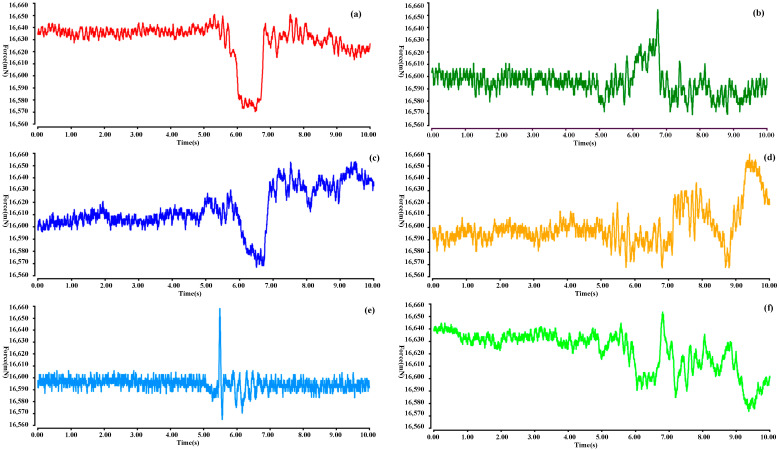
Force curve at the end of six electric cylinders during docking, and figures (**a**–**f**) represent No. 1–6 electric cylinders.

**Table 1 sensors-22-00770-t001:** 6-DOF platform parameter table.

Description	Value
Up and down calibration range of electric Median length of electric cylinder	464–614 mm 539 mm
Moving platform radius	127.5 mm
Static platform radius	162.5 mm

**Table 2 sensors-22-00770-t002:** Electronic system parameter list.

Description	Model/Value
The controller	Beckhoff
Communication protocol of the terminal module	EtherCAT
Laser ranging sensor	200–1000 mm
The torque motor	42–80 Nm
Laser locking sensor	±80 mm

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
