# Peer review of "A Docking Mechanism Based on a Stewart Platform and Its Tracking Control Based on Information Fusion Algorithm"

_sensors, 2022, doi:10.3390/s22030770_

Round 1
Reviewer 1 Report
- I am not very sure that this study falls into the scope of sensors.
- As discussed in the introduction, the six-degree-of-freedom platform has been widely used in the research of docking institution, such as aerospace, mobile radar antenna. Then what’s the difficulty and difference between the aforementioned areas and unmanned vehicles? It is suggested to summarize the contribution of this study more clearly, not just introduce what was done by the authors.
- 9-Fig.14 are not clear enough. The size font is too small to see.
- It seems that the authors just used some mature methods directly to realize the docking function without considering any practical difficulties or model uncertainties. I cannot find any contribution in methods, application or others.
Reviewer 2 Report
The docking mechanism designed in this paper is used in the active mechanism to realize a six degree of freedom platform with higher flexibility, controllability and docking accuracy, so as to make the docking mechanism more flexible in mechanical structure.
In general, authors present a docking mechanism based on a Stewart platform and its tracking control. Authors should consider the following comments to clarify the main contributions of their paper.
1.- In the abstract, authors say “of active mechanism and locking mechanism, The information fusion control algorithm”, it should be “of active mechanism and locking mechanism. The information fusion control algorithm”.
2.- In the page 3, in the introduction, authors say “Literature[28-30] studied the nonlinearity, uncertainty and coupling of space electromagnetic docking, applied feed-back linearization and robust H∞ control technology, and adopted inner and outer loop control strategies.”, in this text, authors should include references [a]-[e] which also study the feedback linearization, and inner and outer loop control strategies.
[a] Adapting H-Infinity Controller for the Desired Reference Tracking of the Sphere Position in the Maglev Process, Information Sciences, Vol. 569, 669-686, 2021.
[b] PI-Type Controllers and Σ–Δ Modulation for Saturated DC-DC Buck Power Converters, IEEE Access, Vol. 9, pp. 20346-20357, 2021.
[c] Stabilization of Two Electricity Generators, Complexity, Vol. 2020, pp. 1-13, 2020.
[d] PD Control Compensation Based on a Cascade Neural Network Applied to a Robot Manipulator, Frontiers in Neurorobotics, Vol. 14, 2020.
[e] Sensorless tracking control for a full-bridge Buck inverter-DC motor system: Passivity and flatness-based design, IEEE Access, Vol. 9, pp. 132191-132204, 2021.
3.- In the pages 8, 9, in the equations (1)-(4), authors should clarify if there is a special reason to consider a image model instead of a dynamic model of their mechanism.
4.- In the page 9, in the equation (5), authors should clarify if the translation and rotation are required in their mechanism.
5.- In the pages 9, 10, in the equations (6)-(10), authors should clarify the main goal of their geometric relationship of their mechanism.
6.- In the page 11, in the equation (12), authors should clarify if this equation describes the control of the mechanism.
Author Response
Please see the attachmen

Reviewer 3 Report
In introduction, the authors presented a lot of work about spacecraft. However, I would like to read how is your work similar to the related research, why is your work better and how is it different from the related research? Please specify the concrete contributions of your paper.
Values in Table 1 indicate that the whole system is really small, is that how it is in the real life? Would the dimensions affect the success rate?
You only performed one experiment? Is this sufficient?
You moved the platform 75 mm in z-direction from which initial position? Does your system work only if the platform is very close to the docking station? How is this connected with the graphs in figure 11 - it seems that distance in z-direction is around 0.3 m?
I also don't quite understand how did you simulate vision sensors (line 369 The relative pose between coordinate systems is used to sim-
ulate the observation of vision and laser sensor)?How is it clear that visual sensors are not sufficient?
If you provided some ground truth values, maybe all of the measurement graphs would be easier to understand.
What is the meaning of graphs shown in Figure 14?
Can you give an example of using the proposed system? What would be the real life scenario problems, for example visibility in the dark etc.
I have some recommendations regarding misspelling, writing and text formatting in general. In line 224 something is not right with the sentences, please reread it and correct it.
The whole section 2.2 is weirdly written. For example, line 239: You need to rotate the docking... It seems like it was copy-pasted from some tutorial or pseudo code.
Page 242, in the flow chart: Whether the centers of the two institutions are aligned - is the institutions the right word?
Line 271, in this equation you used x symbol - did you mean to use \cdot instead?
Please reread your paper and try to find more grammar and spelling mistakes.
Equation (9), why don't you use \Phi, instead of f?
Line 411 - the sentence has repeating of the words.
Line 440 - You left some of the instructions from the template.
Reviewer 4 Report
In line 13, watch the capital letter or the punctuation.
Considering the international character of the journal, the geographic distinction of the literature between lines 47 and 100 should not be a criterium of organization of the text. In this context, the term “foreign research” in line 69 is improper. Please, reorganize lines 47 to 100 based on the relation of the literature with the present paper and not with the geographic location of the researchers.
In Figure 4 make the sentences fit inside the question blocks.
The other parts of the paper are very well written.
Round 2
Reviewer 1 Report
- The contribution should be consistent in the abstract, introduction, main bodies and conclusion. From the lines 14-16 in the abstract, it seems that the main contribution of this study is to present a control strategy. This is not consistent with the answer to Q1 in the last review.
- As in the answer to Q2 in the last review, the author claims that UGV faces more complex and harsh working conditions. Then it is suggested to give out the specific information about these difficulties, and accordingly how these difficulties can be dealt with by the proposed method? Moreover, during the simulation study, how and what of these complex and harsh working conditions are considered? Otherwise, it is hardly to say that the proposed method can deal with these problems successfully and effectively.
- During the simulation study, the author just listed many figures, but the analysis is not enough. For example, why there exist high frequency oscillations in Fig. 9. This is reasonable or not? How can we conclude that the proposed method can deal with complex and harsh conditions? Is there any direct relationship or obvious results?
- In the simulation part, more details about the simulation model should be provided. For example, how about the dynamics of UGV, 6 DOF platform?
Reviewer 2 Report
Authors attended the suggestions of the reviewer. Thus, this paper should be accepted as it is.
Author Response
Thank you very much for receiving this article.
Reviewer 3 Report
The authors have addressed and corrected all comments and suggestions, thus I recommend accepting the paper.
Author Response

(The authors gave the same response as above.)
